# Intercollicular commissural connections refine the representation of sound frequency and level in the auditory midbrain

Llwyd David Orton, Adrian Rees*

Institute of Neuroscience, Newcastle University, Newcastle upon Tyne, United Kingdom

**Abstract** Connections unifying hemispheric sensory representations of vision and touch occur in cortex, but for hearing, commissural connections earlier in the pathway may be important. The brainstem auditory pathways course bilaterally to the inferior colliculi (ICs). Each IC represents one side of auditory space but they are interconnected by a commissure. By deactivating one IC in guinea pig with cooling or microdialysis of procaine, and recording neural activity to sound in the other, we found that commissural input influences fundamental aspects of auditory processing. The areas of nonV frequency response areas (FRAs) were modulated, but the areas of almost all V-shaped FRAs were not. The supra-threshold sensitivity of rate level functions decreased during deactivation and the ability to signal changes in sound level was decremented. This commissural enhancement suggests the ICs should be viewed as a single entity in which the representation of sound in each is governed by the other.

## Introduction

The bilateral organization of the pathways mediating the orienting senses is optimized to detect stimuli occurring in the right and left halves of space relative to the body's midline. Generation of a unified representation of sensory space requires a functional interaction between the two sides of the pathway, which, in the case of vision and touch, occurs in the cerebral cortex predominantly via the corpus callosum. Hearing, however, is distinct amongst the senses in having extensive pre-thalamic processing in the brainstem. Here we report evidence demonstrating that the fundamental properties of sound frequency and level are influenced by extensive interactions between the left and right sides of the pathway at this more peripheral level.

The sub-thalamic auditory system consists of chains of nuclei in the brainstem giving rise to multiple parallel pathways represented in mirror image about the midline. Processing in each side is dominated by the respective contralateral sound field. Each limb of the pathway culminates in the midbrain in the respective left or right inferior colliculi (ICs) whose outputs provide the primary ascending input to the auditory thalamus and cortex on each side. While the left and right pathways are distinct, they involve crossed connections at several levels (*Moore and Osen, 1979*; *Glendenning and Masterton, 1983*; *Glendenning et al., 1985*; *Oliver, 2000*; *Cant and Benson, 2003*). Some of these crossed connections, such as those between the cochlear nuclei and the superior olivary complex, occur between different levels in the hierarchy and mediate interactions that extract the interaural disparities underlying sound localization. Such connections have been termed the acoustic chiasm (*Glendenning et al., 1985*). In addition, there are commissural connections that interconnect mirror opposite nuclei. These are found as early in the pathway as the cochlear nuclei (*Adams and Warr, 1976*; *Cant and Gaston, 1982*), while, in the midbrain, the commissure of Probst connects the dorsal nuclei of the lateral lemniscus (DNLLs) (*Goldberg and Moore, 1967*; *Hutson et al., 1991*). But the most prominent

*For correspondence: adrian.
rees@ncl.ac.uk

**Competing interests:** The
authors declare that no
competing interests exist.

**Reviewing editor**: Ranulfo
Romo, Universidad Nacional
Autonoma de Mexico, Mexico

**eLife digest** The bilateral arrangement of our eyes and ears enables us to receive information from both sides of our body. This information is conveyed via various sensory pathways that take different routes through the brain to culminate in the cerebral hemispheres. The information is then processed in the brain's outer layer, which is called the cortex.

In the visual system, information from both eyes is kept separate until it reaches the cortex. A similar arrangement exists for touch. However, hearing is unusual among our senses in that sounds undergo much more processing in the brainstem, which is located at the base of the brain, than other types of stimuli. Orton and Rees now show that, in contrast to vision and touch, information about sounds occurring to our left or right is refined by interactions between the two sides of the midbrain.

To test for sideward interactions between the two limbs of the auditory pathway, electrodes were lowered into the brains of anesthetized guinea pigs so that neuronal responses to tones could be recorded. The electrodes were placed in the region of the midbrain that contains two structures called the inferior colliculi (meaning 'lower hills' in Latin). Each inferior colliculus predominantly receives inputs from the opposite ear. However, recordings made in one colliculus when the other was deactivated revealed that one colliculus normally alters the response of the other. This shows that there is an important sideward interaction between the two halves of the auditory pathway in the midbrain that refines how fundamental aspects of sound, such as its frequency and intensity, are processed.

This represents a marked departure from our previous understanding of auditory processing in the mammalian brain, and opens up new lines of investigation into the functioning of the auditory system in health and disease.

of these connections is the commissure of the inferior colliculus (CoIC) (*Beyerl, 1978*; *Adams, 1980*; *Aitkin and Phillips, 1984*; *Coleman and Clerici, 1987*).

The CoIC interconnects the ICs in a systematic fashion that mirrors the spatial representation of frequency (tonotopicity) in these nuclei (*Adams, 1980*; *González-Hernández et al., 1986*; *Saldaña and Merchán, 1992*; *Malmierca et al., 1995*, *2009*). These fibers are predominantly excitatory although a significant minority (~10–20%) are GABAergic (*González-Hernández et al., 1996*; *Hernández et al., 2006*; *Nakamoto et al., 2013*). This commissure is the final point of interaction between the two limbs of the ascending auditory pathway prior to the auditory cortex.

The preponderance of these commissural inputs and the importance of interaural comparison in the analysis of sound suggest that the two ICs may operate in a more cooperative manner than has hitherto been considered. Recent in vivo studies in the IC in response to sound stimuli that were varied in azimuthal location have led to the speculation that the CoIC may mediate binaural processing between the ICs (*Malmierca et al., 2005*; *Li and Pollak, 2013*; *Ono and Oliver, 2014*). Rather than each IC being an independent entity, we hypothesized that functionally the ICs should be considered as two halves of a whole. This commissural influence might manifest itself in one of two ways, either as a global regulation of firing, or as differential changes in the firing patterns of different unit types, indicating more functionally specific control.

To test these hypotheses we have examined the extent to which two fundamental aspects of sound processing, neuronal responses to frequency and level, in one IC are governed by the commissural connections between them. We have recorded response properties of neurons in one IC of guinea pig while deactivating the other IC using two different methods, cooling using a cryoloop (*Lomber et al., 1999*; *Orton et al., 2012*) or microdialysis of the short acting local anesthetic procaine (*Boehnke and Rasmusson, 2001*). These methods enabled us to address the causal impact of these connections on individual neurons.

Our findings show that intercollicular commissural connections regulate firing and are a major determinant of the receptive fields of IC neurons. Intercollicular processing increases the heterogeneity of receptive field types with implications for enhancing the perception of frequency, and the ability of IC neurons to convey changes in sound level. Our results imply that the analysis of frequency and level, both fundamental components of hearing, and generally considered to be monaural

processes, depend on bilateral interactions between the ICs. These findings demonstrate the two sides of the auditory midbrain operate collectively rather than independently of one another.

## Results

To investigate the influence of the CoIC on auditory processing we made extracellular recordings from 125 single units in the right IC in response to diotic pure tones to generate frequency response areas (FRAs) (*LeBeau et al., 2001*) and rate level functions (RLFs) (*Rees and Palmer, 1988*). Recordings were made before, during and after deactivation of the left IC using either cryoloop cooling (*Orton et al., 2012*) or microdialysis of procaine (MDP) in anesthetized adult guinea pigs. We have demonstrated that the cryoloop in our implementation abolished or reduced firing rates in the dorsal half of one IC (to the depth at which frequencies of up to ~8 kHz are represented in the tonotopic sequence) while leaving ventral regions less affected. Cooling did not spread to the contralateral IC and did not modulate the afferent volley to the contralateral IC, showing that changes imparted by the cooling protocol in the contralateral IC were of commissural origin (*Orton et al., 2012*). In this study we first sought to establish the use of MDP to replicate the effects of cooling, in deactivating the left IC.

### Microdialysis of procaine caused rapid, reversible neural deactivation in one IC

We first verified the use of MDP in our preparation by recording multi-unit activity in the left IC adjacent to the microdialysis probe, before, during and after procaine dialysis. The schematic coronal section through the preparation in *Figure 1A* shows the placement of the microdialysis probe in the center of the exposed IC with a microelectrode inserted adjacent to the probe.

Repeated PSTHs showed a stable firing rate with aCSF flowing through the microdialysis probe (*Figure 1B*), and a rapid reduction in firing rate when procaine was dialyzed. Firing rate recovered over the course of 20 min following aCSF washout through the probe. *Figure 1C* shows the multi-unit FRA in the control condition. MDP abolished firing at almost all frequencies and levels of sound stimulation. The reduction in firing rate was similar across the entire response area. After 20 min of aCSF washout the response area returned to control. Insertion of the microdialysis probe into the left IC produced negligible trauma. The area of the IC and the area of the lesion caused by the probe were measured in Nissl stained sections. Lesions comprised just 1.49% (±0.31) of the area of the IC.

While procaine has a high pK$_a$, and consequently has low efficacy in vivo compared to other sodium channel blockers that are commonly used in deactivation studies (*Truant and Takman, 1959*), recovery from deactivation is faster than with amide local anesthetics such as lidocaine (*Sung and Truant, 1954*) as ester local anesthetics are metabolized in plasma rather than liver (*Covino, 1981*). This avoided the spread of procaine deactivation to other areas of the brain as its metabolism begins in situ. Procaine has no effect of resting membrane potential (*Taylor, 1959*) or spike amplitude (*Butterworth and Cole, 1990*). Microdialysis of procaine rapidly deactivated neurons without producing the mechanical disturbances associated with pressure injection (*Malmierca et al., 2003*, *2005*).

### Firing rate, but not receptive field area, is influenced by the CoIC in V FRAs

Frequency response areas in the IC are diverse; however, they can be grouped as being either V-shaped, with responses similar to those found in the in the auditory nerve, or into a more heterogeneous group collectively termed nonV. The latter are an emergent property of central processing. At least seven response area types can be defined within this group, but they represent points on a continuum rather than discrete classes (*Palmer et al., 2013*). For convenience we will refer to these two classes as V and nonV FRAs, respectively.

We measured the excitatory response area by finding frequency-level combinations within each FRA in which the firing rate exceeded the mean spontaneous firing rate by two standard deviations (*Ingham et al., 2006*). We also measured the number of spikes elicited by the unit in each FRA. These two measures were compared between conditions. Similar effects were found in response to either cooling or MDP deactivation of the contralateral IC. Control units tested with cooling or MDP did not differ in characteristic frequency (CF), FRA area or firing rate. For analysis we therefore pooled these responses.

*Figure 2* shows four examples from the sample of 49 V-shaped FRAs in our dataset. The area of V FRAs did not change in response to either cooling (n = 37) or MDP (n = 12) deactivation of the

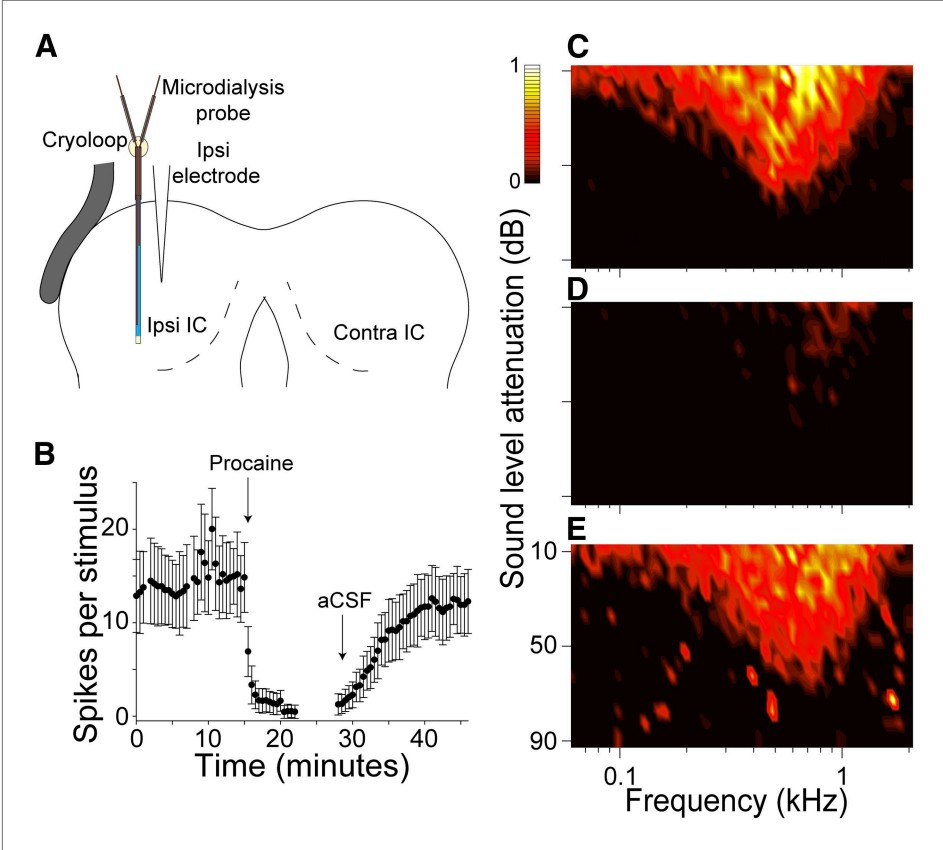

**Figure 1**. Microdialysis of procaine into the IC produced a rapid, reversible deactivation of spiking. (**A**) Schematic coronal image of the setup for recording the effects of procaine on neuronal activity in the left IC. (**B**) Mean (±SD) firing rate of multi-units recorded by an electrode adjacent to the probe, each point represents the average spikes per stimulus value in response to 100 repetitions of a CF tone at 20 dB above threshold. Procaine caused an immediate and persistent deactivation throughout infusion. After switching back to aCSF, firing recovered to near control levels after 20 min. (**C**) Multi-unit FRA recorded from an electrode adjacent to the microdialysis probe. (**D**) Procaine infusion abolished firing at virtually all frequencies and levels. (**E**) Response area shape and firing rate recovered similar to control.

contralateral IC. The absence of changes in FRA area was apparent throughout recovery. Conversely, changes in firing rate within and across the existing response area occurred for most V FRAs in response to cooling and MDP deactivation. Both increases and decreases in firing rate were seen in the population and these recovered towards control levels after deactivation was stopped. With a 20% change in firing rate as a criterion of change (*LeBeau et al., 2001*, *Burger and Pollak, 2001*; *Park et al., 2008*), 14% (7/49) V FRAs decreased in firing rate while 35% (17/49) increased.

## Firing rate and area of nonV FRAs are governed by the CoIC

We recorded examples of all previously reported nonV FRA response types, and in *Figure 3* show four examples from the 50 in our dataset. In contrast to V FRAs, FRAs in the nonV group showed dramatic changes in area following deactivation of the other IC (*Figure 4C,D*). For example, the neuron whose *low-tilt* FRA is shown in *Figure 3A* declined in both spontaneous and auditory driven firing rate and its FRA reduced in area by 52% (*Figure 3B*). The *narrow* FRA type in *Figure 3D* reduced in area by 86% (*Figure 3E*) and the *closed* FRA in *Figure 3G* expanded in area by 234% (*Figure 3H*) and showed an elevation in firing rate for frequencies centered on its CF, the frequency to which it was most sensitive. The *broad* FRA in *Figure 3J* reduced in area by 40% (*Figure 3K*). All of these changes reversed to near control levels on recovery (*Figure 3C,F,I,L*). Quantitative analysis found that the firing rate of 34% (17/50) of nonV FRAs decreased in firing rate while 28% (14/50) increased.

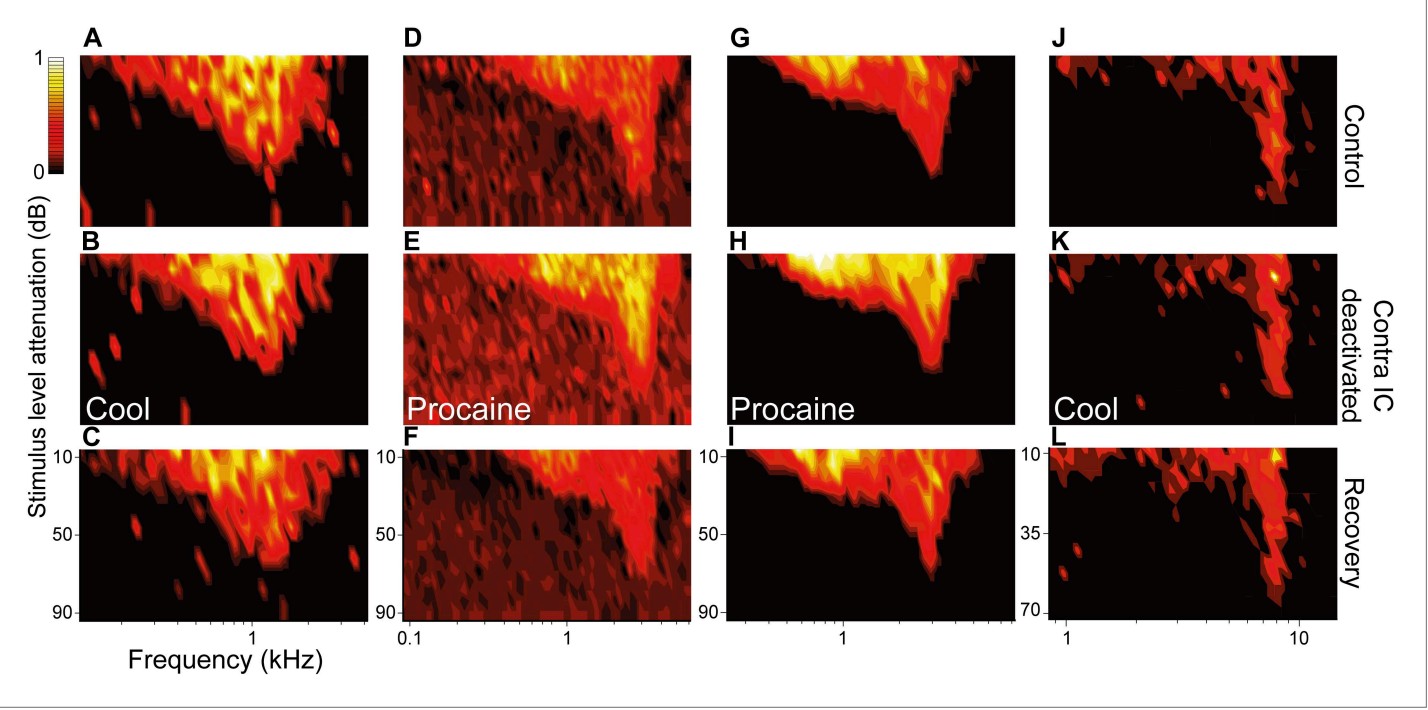

**Figure 2**. Deactivation of the contralateral IC influenced the firing rate of IC neurons with V responses but had little effect on their shape or area. Columns (**A**, **D**, **G**, **J**) show four examples of IC units in control (top), deactivated (middle) and recovery (bottom) conditions with responses for each unit normalized to the maximum firing rate across conditions. While firing rate changes within V FRAs were common, the tuning and shape of these FRAs were unaffected by deactivation of the contralateral IC by either cooling or MDP. All changes in firing rate during deactivation (**B**, **E**, **H**, **K**) recovered following cessation of cooling or procaine infusion (**C**, **F**, **I**, **L**).

## FRA type heterogeneity is increased by commissural processing

Plotting the spikes within each FRA in the deactivated condition as a function of FRA spikes in the control condition revealed changes in all FRA types (**Figure 4A**). We quantified the change in spiking in both V and nonV FRAs by calculating a modulation index:

$$MI = (deactivated - control)/(deactivated + control).$$

The distributions of change in firing rate for both V and nonV FRAs (**Figure 4B**) were similar (two sample Kolmogorov-Smirnov test: D(97) = 0.20; p = 0.26).

There was a difference between the change in area for the two FRA classes. The distribution of V FRA areas was almost unchanged during deactivation (**Figure 4C**), but the range of changes in nonV FRAs was more than double that of V FRAs (**Figure 4D**). Comparison of FRA area change between V and nonV FRAs showed that nonV FRAs were modulated more than V (D(97) = 0.32; p = 0.009).

## Commissural input creates non-monotonic responses

Having revealed a distinction between the effect of commissural inputs on V and nonV frequency receptive fields, we next sought to investigate how changes in response type would influence the encoding of sound level. To this end, we generated RLFs for single units by recording responses to multiple presentations of pure tones at the unit CF at several sound levels. Effectively, this is a measure of a neuron's response to stimuli on a vertical line transecting the FRA at CF. Each RLF was classified into one of five previously reported types (**Rees and Palmer, 1988**; **Winter et al., 1990**). Non-monotonic RLFs were defined using the criteria of **Rees and Palmer (1988)**, where a reduction in firing of 25% or more was required at a sound level above that which produced maximal firing. Non-monotonic RLFs comprised 12 of the 38 neurons recorded. A range of FRA types can give rise to non-monotonic RLFs. While examples of all RLF types showed firing rate changes during contralateral IC deactivation, only

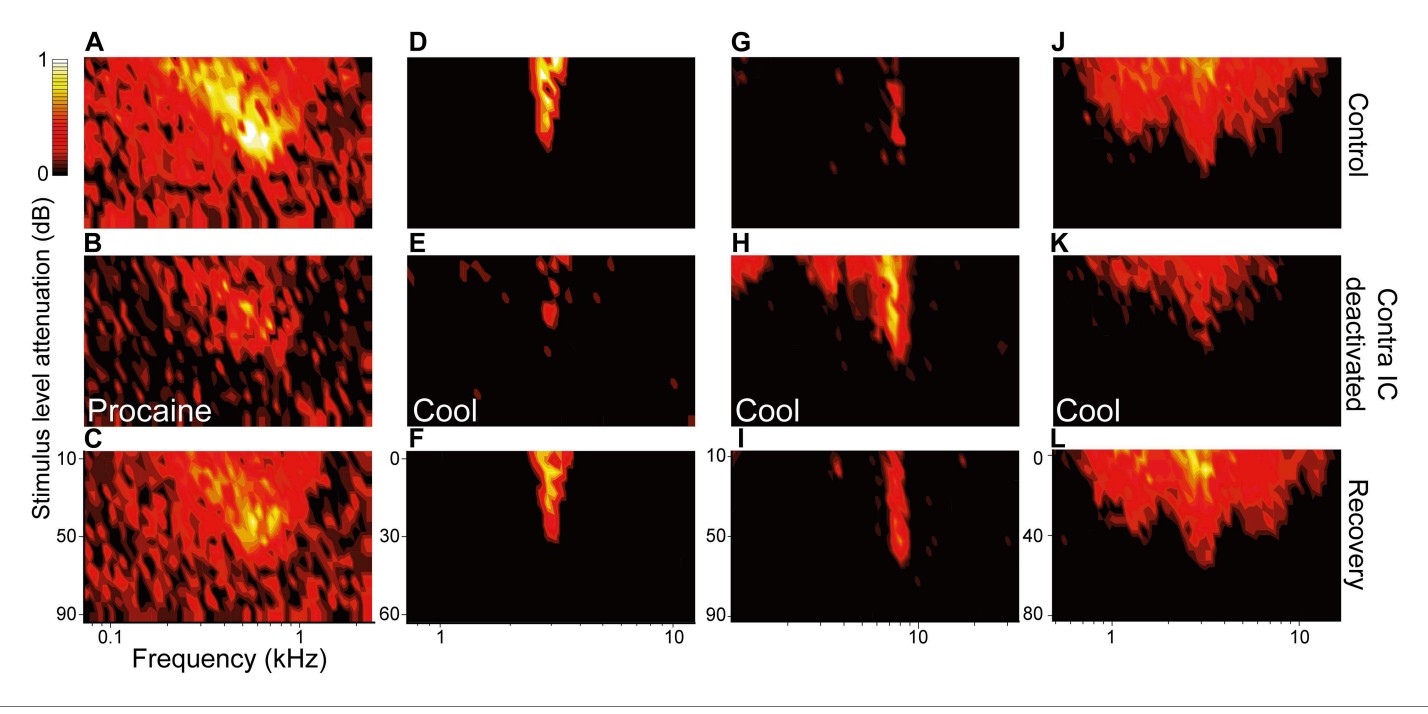

**Figure 3**. Deactivation of the contralateral IC influenced firing rate as well as the receptive field area and shape for nonV responses. (**A**) Low tilt FRA with high spontaneous rate. (**B**) Procaine in the contralateral IC caused a reduction in both driven and spontaneous firing and FRA area. (**D**) Narrow FRA which (**E**) reduced in area on contralateral cooling but fired sporadically over a wider frequency range. (**G**) Closed FRA which (**H**) expanded in area and increased in firing rate. (**J**) Broad FRA which (**K**) reduced in rate and area on contralateral cooling. (**C**, **F**, **I**, **L**) All changes in firing rate and response area reversed on recovery.

non-monotonic RLFs changed shape. Four of the 12 non-monotonic RLFs became monotonic (*Figure 5A,C*), four became straight (*Figure 5B*), while the remainder remained non-monotonic.

To analyze these effects we constructed a 5 × 2 contingency table with each row comprising one of the five RLF types and each column the sum of a binary classifier applied to each RLF as to whether it changed in type (1) or maintained the same type (0) on contralateral deactivation. A Fisher's exact test found that the probability of the observed distribution occurring or one more extreme, was 0.0002. Thus, non-monotonic RLFs in IC are a distinct type of RLF whose shape can be formed by CoIC processing.

## Commissural input enhances the supra-threshold representation of sound level

We assessed the impact commissural deactivation has on the supra-threshold encoding of changes in sound level by calculating the half maximum firing rate for RLFs in each condition (*Figure 5A–C*). In the majority of units, deactivation shifted RLFs so that a higher sound level was required to evoke the half maximum firing rate than in the control condition (*Figure 5D–F*). On recovery, the half maximum firing rate of these RLFs returned to values similar to the control condition.

In the population of RLFs (*Figure 5G*) the median sound level required to elicit half maximal firing increased from 50.9 dB sound pressure level (SPL) (95% CI: 46.1 56.2) to 59.3 dB SPL (95% CI: 55.2 65.0) on deactivation. On recovery, the median half maximal level of the distribution returned to 51.3 dB SPL (95% CI: 47.5 61.1). The increase in sound level required to reach half maximal firing rate during deactivation indicated that CoIC rescaling improved supra-threshold coding of sound level (Friedman repeated measures ANOVA on ranks: $\chi^2(2) = 12.72$; $p = 0.001$). *Post hoc* analysis found a higher sound level was required to elicit half maximum firing during contralateral IC deactivation than during control recordings ($Z = -3.94$; $p = 0.00008$). There was no difference between the level required to evoke half maximal firing in the control and recovery conditions ($Z = -1.41$; $p = 0.15$).

If this effect was due to a shift in the entire RLF to a higher sound level, changes in threshold would correlate with changes in the half maximum firing point on the RLF. To test this we performed a

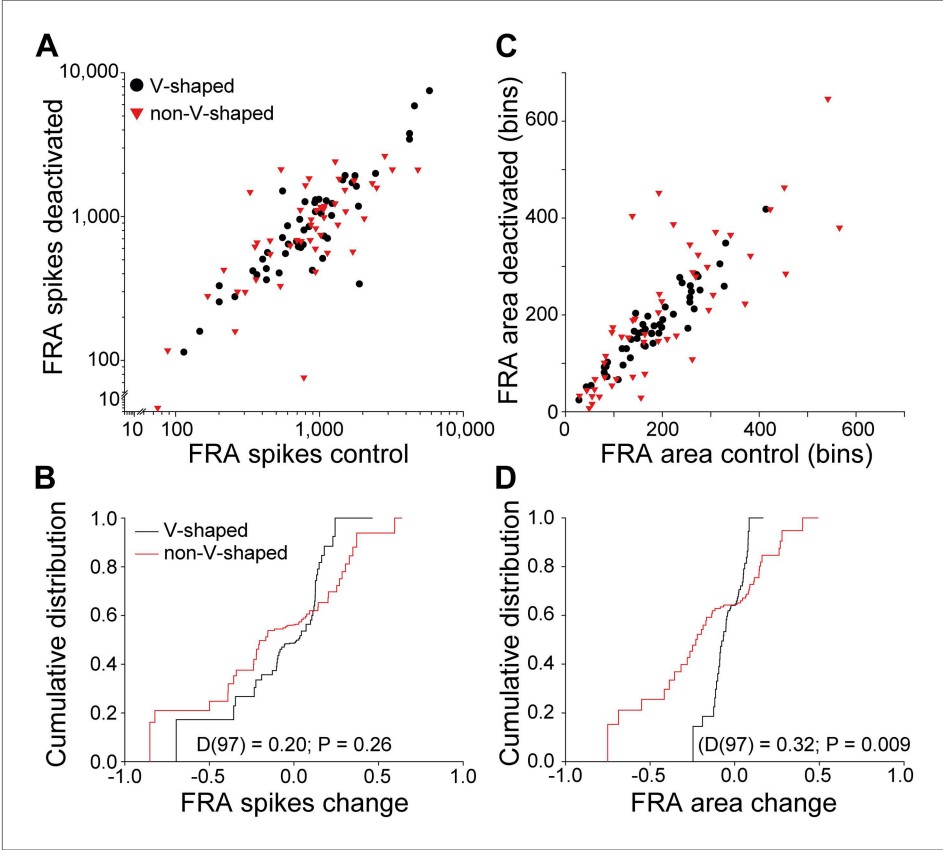

**Figure 4**. Differential changes in firing rate and receptive field area by FRA class. (**A**) Total spikes in FRA on deactivation of the contralateral IC as a function of control value. A wide range of changes in firing rate were noted for both V and nonV receptive field classes with deactivation. (**B**) Cumulative distribution of change index (see text) for total spikes showing a similar range and distribution for V and nonV FRAs. (**C**) Receptive field area on deactivation as a function of control. (**D**) Cumulative distributions of change index of FRA area. The range of area change for nonV units was three times greater than for those with V response areas.

correlation analysis between the change in half maximum firing and the change in statistical threshold (**Ingham et al., 2006**) for neurons for which both FRAs and RLFs were collected (**Figure 5H**). There was no correlation between the change in threshold and the change in half maximum firing ($r_s(14) = 0.20$; p = 0.45). This demonstrates that the predominant effect of removing CoIC input was to rescale suprathreshold responses, consistent with the operation of a gain control of neuronal output.

## Commissural input improves sound level discrimination

An important question is whether the changes in RLFs during deactivation altered the ability of units to signal changes in sound level. To investigate this we devised a measure to assess the discriminability of sound level changes for each RLF. We constructed an ROC curve from the responses to adjacent stimuli in each RLF in each condition (**Figure 6A**) and calculated the area under each curve for each pair (**Figure 6B**) before summing the absolute deviation of each receiver operating characteristic (ROC) curve from 0.5. This produced a 'Discriminability Index' (DI) which quantified how well changes in sound level could be decoded by an ideal observer from changes in firing rate along each RLF.

*Figure 6C* shows the RLFs from a neuron with a monotonic saturating response throughout the three conditions in the experimental paradigm. The DI of this unit in the control condition was 1.5 (*Figure 6D*). The firing rate and slopes of the RLF were reduced on contralateral IC deactivation leading to a fall in DI to 1.2. On recovery, the RLF returned to near control values and the DI increased to 1.4.

For the 38 units, the population DIs in the control condition (median = 1.72; 95% CI: 1.52 1.99) were reduced during contralateral IC deactivation (median = 1.54; 95% CI: 1.37 1.68) although some units

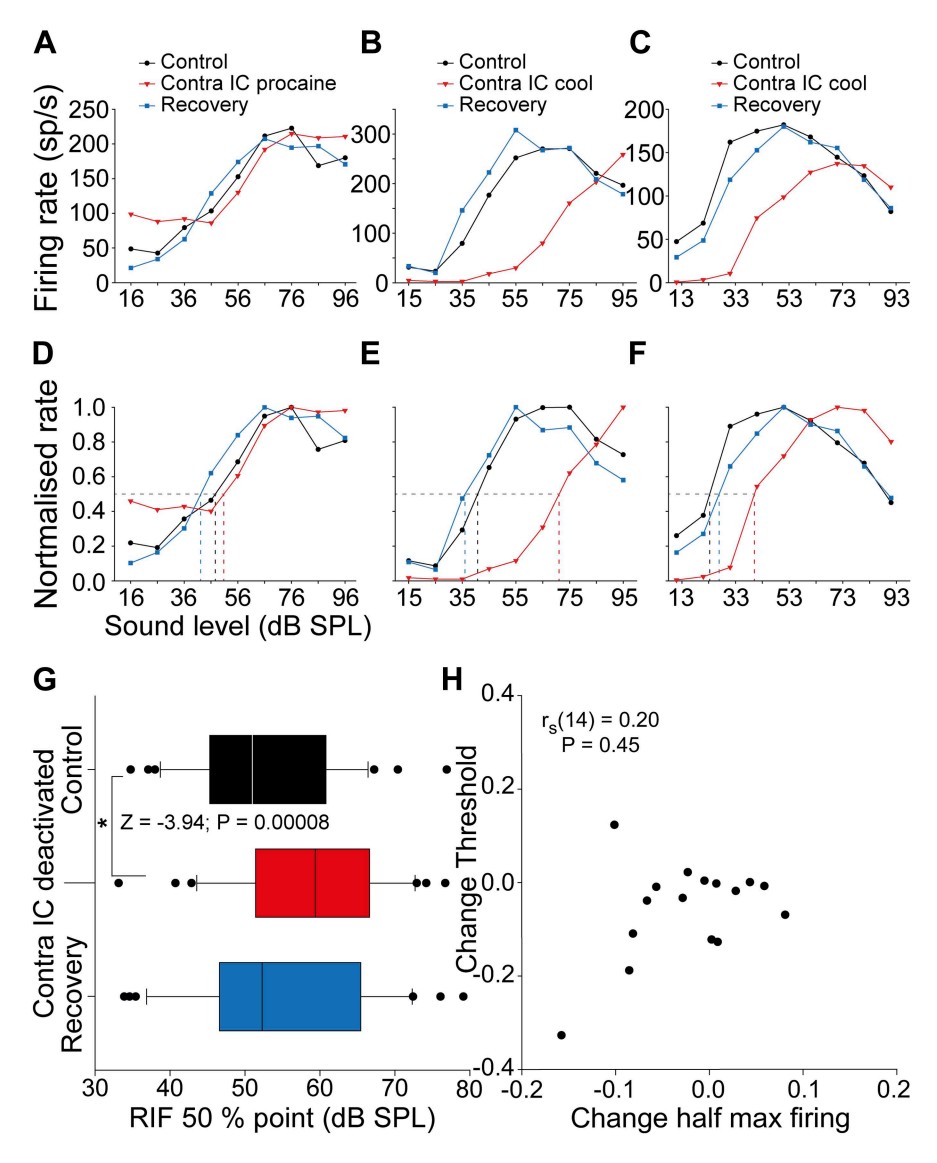

**Figure 5**. Changes in the shape and half maximum firing rate of RLFs on deactivation of the contralateral IC. (**A–C**) Control RLFs (black) with non-monotonic response functions with increasing sound level. With deactivation (red) RLFs became more monotonic and in some (**B** and **C**) the curves shifted to higher sound levels. These changes reversed on recovery (blue). (**D–F**) Normalizing the functions in the top row shows the sound level required to elicit half maximal firing increased during deactivation. (**G**) The population of half maximum firing rates increased on deactivation of the contralateral IC. (**H**) Changes in half maximum firing rates did not correlate with changes in threshold, indicating that the dominant effect of commissural input on firing rate was at supra-threshold levels.

did show an elevation (*Figure 6E*). Deactivating the contralateral IC reduced the distribution of DIs (Wilcoxon signed rank test: Z = −2.18; p = 0.028).

Such changes demonstrate that the supra-threshold gain control which the CoIC exerts over IC responses enhances the ability of IC neurons to discriminate changes in sound level.

## Discussion

Our results show that neuronal responses and the formation of receptive fields in the auditory midbrain depend on interactions between the bilateral limbs of the auditory pathway. Specifically, activity mediated by commissural fibers connecting the ICs influences the responses of almost all the

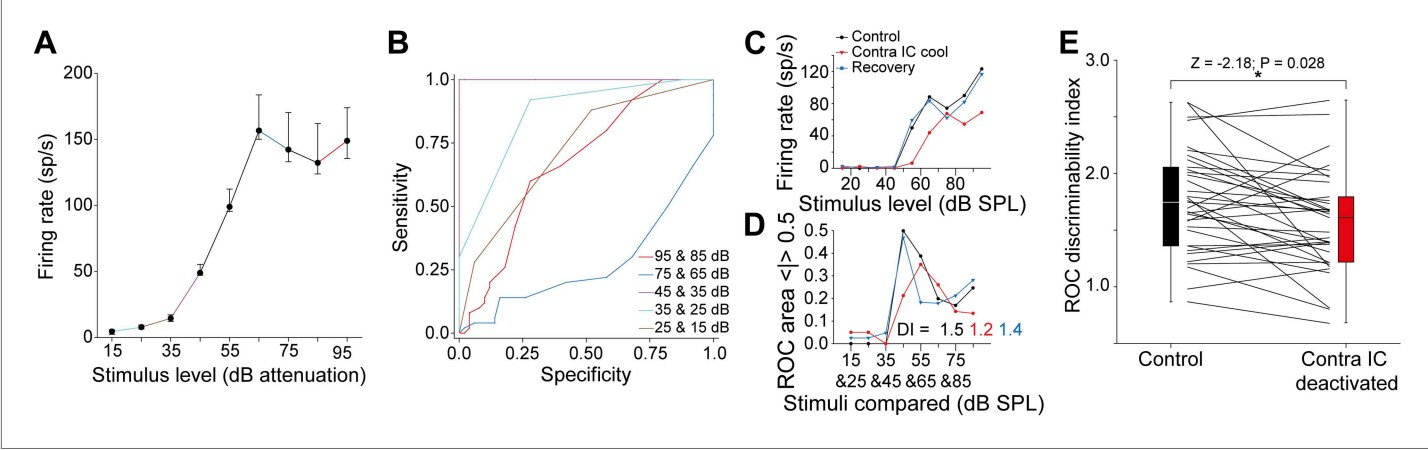

**Figure 6**. Deactivation of the contralateral IC reduced sound level discriminability in the IC. (**A**) Control RLF of an IC neuron (median ± IQR). (**B**) ROC pairwise analysis of the responses in (**A**) with colors of ROC curves matching the corresponding paired values in (**A**). (**C**) An example RLF (black) which showed a reversible reduction in rate and slope on deactivation (red) that recovered to control values (blue). (**D**) A 'discriminability index' (DI) was calculated from the change in area under the ROC (see text) for each adjacent stimulus pair in each condition. The reduction in rate and slope in (**C**) on deactivation resulted in a reduction in DI from 1.5 to 1.2 with recovery to 1.4. (**E**) The median DI for the population of RLFs declined on deactivation of the contralateral IC, indicating a reduction in discriminability of sound level on in the absence of commissural input.

IC neurons we recorded in at least one of four ways. The majority of neurons in our sample showed changes in firing rate following deactivation of the other IC (*Figure 4A,B*). Neurons with nonV frequency response areas also showed changes in receptive field shape, often (but not exclusively) becoming more V-like in the absence of commissural input (*Figure 4C,D*). We observed that deactivating inputs from the opposite IC reduced supra-threshold firing rates to sounds of varying level (*Figure 5*). Finally, deactivation reduced the ability of IC neurons to signal changes in sound level by changes in firing rate (*Figure 6*). Similar changes were observed whether the IC was deactivated by cooling or MDP.

Previous studies aimed at deactivating one colliculus were limited by their methodology so only a small number of units were recorded (*Malmierca et al., 2003*, *2005*). Although the responses of individual units in those studies are consistent with our data, the population based patterns of change we reveal here were not apparent. The two neural deactivation methods we have applied in this study have the advantages of 1) not needing direct interference with the preparation during deactivation, and 2) rapid recovery from deactivation. We have shown that cooling as a method of deactivation is restricted to the IC when cooling cycle durations are less than 25 min (*Orton et al., 2012*). Under these conditions, deeper structures, such as the DNLL, appear unaffected. We have also demonstrated reversible deactivation with a second method, MDP with effects consistent with cooling (*Figure 1*). The methods are complementary in that cooling effects are more extensive within the IC, and consequently produce more substantial changes in the other IC but MDP provides greater control over the extent of the deactivation, albeit with generally more modest effects. The similarity of the datasets using the two methods also gives confidence that they reflect the consequences of deactivating the contralateral IC. It is not impossible that some of the consequences of deactivation are mediated via descending pathways originating in the IC to lower brainstem structures, but this seems unlikely. Recordings of evoked potentials from the recorded IC while cooling the other show significant changes at latencies consistent with processing in the IC, but little or no effect on the incoming afferent volley to the IC (*Orton et al., 2012*).

The anatomical basis for these physiological observations is the CoIC, which tracer experiments demonstrate interconnect the frequency-band laminae in the IC (*González-Hernández et al., 1986*; *Saldaña and Merchán, 1992*; *Malmierca et al., 1995*, *2009*). Immunohistochemical studies suggest that the majority of commissural fibers are excitatory, with only 10–20% labelling for the inhibitory neurotransmitter GABA (*González-Hernández et al., 1996*; *Hernández et al., 2006*; *Nakamoto et al., 2013*). The changes in response properties we observed are consistent with commissural input exerting both excitatory and inhibitory influences, with much greater inhibition than would be

predicted from the ratio of excitatory to inhibitory fibers. This suggests that within the contralateral IC, excitatory commissural fibers can exert their effects di- or poly-synaptically via inhibitory neurons in the target IC. Evidence supporting this idea comes from intracellular recordings in IC slices in which electrical stimulation of the CoIC revealed inhibitory currents that were reduced by glutamatergic blockade consistent with di-synaptic inhibition (*Smith, 1992*; *Moore et al., 1998*).

Of particular interest is the effect of commissural inputs on the frequency receptive fields of IC neurons. Of the two major classes, V FRAs have many features in common with the responses of the primary fibers that emerge from the cochlea. However, unlike primary fibers, these neurons do receive inhibitory inputs, but this inhibition does not change receptive field shape (*Egorova et al., 2001*; *Alkhatib et al., 2006*; *Palmer et al., 2013*). The increases in firing rate seen here within V response areas are similar to those obtained by iontophoretic blockade of GABA and glycine receptors in IC (*Vater et al., 1992*; *Yang et al., 1992*; *LeBeau et al., 2001*). Commissural input serves to rescale the output of neurons with V-shaped responses areas by modifying the balance of local excitation and inhibition without changing area of the receptive field (*Figure 2*).

In contrast, nonV receptive fields are more heterogeneous in appearance, and although different classes can be identified, many appear to be intermediate types supporting the existence of continua rather than discrete groups (*Palmer et al., 2013*). Nevertheless all nonV responses are similar in appearing to be shaped from V responses by excitatory and/or inhibitory inputs. Whether this shaping occurs in the inferior colliculus or is inherited from earlier stages of the auditory pathway is still contentious. Both the firing rates and response areas of nonV units were modified by deactivation of the contralateral IC (*Figure 3* & *Figure 4B,D*). Often, but not always, nonV FRAs became more V-shaped (*Figure 3*). The changes in receptive field shape and firing rate we observed on deactivation of the other IC show that it is directly or indirectly a source of the inhibitory input that generates these receptive fields.

Increasing diversity of spectral responses is a feature of the ascending auditory pathway that presumably reflects the increasing complexity of analyses performed at each level in the process of signal extraction. There is a greater diversity of FRA types in midbrain than in the brainstem (*Evans and Nelson, 1973*; *Hernández et al., 2005*; *Ingham et al., 2006*; *Palmer et al., 2013*) and other types not described in sub-thalamic nuclei are reported in auditory cortex (*Shamma et al., 1993*; *Recanzone et al., 2000*; *Bitterman et al., 2008*; *Bartlett et al., 2011*; *Atencio et al., 2012*; *Grimsley et al., 2012*). The contribution of commissural input to this process has not been previously considered. We have demonstrated that lateral connections between the two sides of the auditory pathway play an important role in shaping response areas in the IC, and thus increasing response-type diversity. It may not be the sole contributor to this process, and the extent of its contribution remains to be determined.

The analysis of input-output functions for sound level shows that commissural input contributes to the non-monotonicity observed in many of these functions (*Figure 5A–C*), as well as to the rescaling of responses that enhances the discriminability of sound level in the responses of IC neurons (*Figure 6E*). The ubiquity of the commissural influence on firing rates supports the contention that the IC exerts an element of gain control on its contralateral counterpart (*Malmierca et al., 2003*, *2005*). Given that a congruent representation of the auditory environment is dependent on combining the contralateral auditory hemifields represented in each IC, the commissure may operate to calibrate one with the other.

Although the differential effects of commissural connections on V and nonV frequency receptive fields and level encoding are clear, how such changes enhance auditory analysis is less so. Stimuli that arrive at each of the two ears are commonly disparate in temporal, level or spectral characteristics. These disparities are fused by central processing into an auditory percept. The construction of left and right auditory hemi-fields is a consequence of monaural and binaural analysis in the brainstem leading to the encoding of sound source location in azimuth and elevation, but our behavioral *sensitivity* to changes in sound position is greatest around the midline where these representations meet (*Mills, 1958*). Furthermore, our ability to separate sound sources, for example to hear out an individual speaker in a noisy social situation, is enhanced by binaural hearing (*Litovsky and McAlpine, 2010*). Commissural modulation of frequency receptive fields in the ICs may assist sound source separation in acoustical cluttered situations. By modulating the spectral area of FRAs in each IC, this processing may increase FRA diversity to overcome correlated noise present in a neural population, and thus improve signal extraction and information content in a neuronal population (*Zohar et al., 2013*).

The IC seems the obvious site at which to interconnect the bilateral limbs of the auditory pathway. It is the first structure where the parallel pathways emerging from the cochlear nucleus (*Harrison and Warr, 1962*; *Osen, 1970*; *Cant and Benson, 2003*) and further elaborated in the brainstem nuclei can be integrated. Thus, the different cues for determining sound location, (interaural time and level disparities and pinna spectral cues) are computed in separate nuclei and are first brought together in the ICs.

Responses to changes in overall sound level at the periphery will always be congruent in the left and right ICs, while changes in azimuthal location will produce changes in firing rate in each IC antipodal to the other. Processing between the ICs likely takes advantage of the symmetrical organization of the CoIC, so that bilateral changes in firing rate in each IC can better represent changes in sound level and location (*Figure 6*).

In conclusion, commissural processing between the ICs exerts control over the firing rate of most IC neurons, and contributes to the creation of receptive fields determining responses to two fundamental parameters of sound: frequency and level. This degree of interaction suggests that each IC should not be considered as independent of the other, but as two halves of a cooperative whole operating in concert to optimize the encoding of sounds.

## Materials and methods

All experiments were performed in accordance with the terms and conditions of a license issued by the UK Home Office under the Animals (Scientific Procedures) Act of 1986, and with the approval of the Local Ethical Review Committee of Newcastle University.

### Animals

Data reported here are from experiments performed on 32 (19 male, 13 female) outbred, pigmented guinea pigs (Cavia porcellus). Animals were housed in pens in a breeding colony on a 12 hr light-dark cycle. The number of animals per pen varied from 1 to 6 depending on the numbers in the colony at any time. The walls of the pens were made from wire mesh so that all guinea pigs in the colony could see and hear other members of the colony. Experimental animals had a median age of 6 months (Inter-quartile range (IQR) = 4 to 7; range = 2 to 13). All animals showed a strong pinna reflex in response to a finger click above the head on the morning of each experiment. The median weight of animals was 796 g (IQR = 678 to 894; range = 460 to 1093).

### Anesthetic protocol

Anesthesia was induced with urethane (0.7–1 g/kg as 20% solution, intraperitoneal injection (i.p.)) and supplemented by Hypnorm (fentanyl citrate 0.315 mg/ml and fluanisone 10 mg/ml; 1 ml/kg, intramuscular injection). Atropine sulphate monohydrate (BDH Chemicals; 0.05 mg/kg, subcutaneous injection) was given to suppress bronchial secretions. Anesthesia was maintained with further doses of Hypnorm (1/3 original dose) as required.

### Surgery

A tracheotomy was performed and a cannula inserted into the stoma and secured with suture. The cannula provided an airway and a means through which the animal could be artificially ventilated if required. Animals were allowed to respire spontaneously. If breathing became labored, the animal was artificially respired with medical air via a modified small animal ventilator (Harvard Apparatus, UK) which maintained end-tidal $CO_2$ at ~5%. Core temperature was monitored with a rectal probe and maintained at 38 ± 1°C with a thermostatically controlled electric blanket (Harvard Apparatus).

Experiments were conducted inside a single walled, sound attenuating room (IAC). Animals were placed in a stereotaxic frame (Kopf, Tujunga, California) in which the standard ear bars were replaced by hollow polymethyl methacrylate conical specula, the apices of which were placed in the auditory meatuses to permit sound delivery. A surgical microscope (Zeiss, UK) was used to visualize the tympanic membranes to check their condition and placement of the specula.

A dorsal midsaggital incision was made along the scalp. The skin was reflected and the tissues overlying the skull were abraded. Two holes were trephined on either side of the midline to expose the occipital lobe covering the left and right ICs and each hole was extended with rongeurs. The dura mater was retracted bilaterally. The cortex overlying the left IC was aspirated with a glass Pasteur pipette attached to a vacuum pump until the dorsal surface of the left IC was visualized.

## The cryoloop and cooling system

The cryoloop and cooling system used in this study were as reported and validated previously (**Lomber et al., 1999**; **Orton et al., 2012**). A cryoloop was constructed from stainless steel tubing. A thermocouple (Omega, UK) was secured to the cryoloop tip to allow monitoring of the cryoloop temperature with a digital thermometer (HH506RA, Omega). A peristaltic pump (Gilson, UK) pumped ethanol at −80°C around a hydraulic system. Regulating the pump speed controlled the flow rate through the system and thereby enabled control of cryoloop temperature. The cryoloop was curved to maximize contact with the dorso-lateral surface of the exposed IC. This cooled areas in which the density of neurons projecting to the contralateral IC is highest (**Saldaña and Merchán, 1992**; **Malmierca et al., 2009**). Cooling cycles were kept as brief as possible to optimize the chance of holding the unit throughout recovery. We have verified this method as a consistent and reliable means to deactivate the left IC by making recordings and temperature measurements in the cooled IC (**Orton et al., 2012**). Cooling reduced firing rates in response to sound stimuli in some cases almost completely and by at least half for units tuned to frequencies up to ~8 kHz. Units tuned to higher frequencies were deactivated less or not at all. This established that cooling deactivated the dorsal half of the left IC. Cooling induced deactivation did not spread to the contralateral IC and did not modulate the afferent volley to the right IC indicating any changes imparted in the right IC were produced by removal of commissural input (**Orton et al., 2012**).

## Microdialysis of procaine

Concentric microdialysis probes were constructed as described previously (**Gartside et al., 1996**). Stainless steel tubing (AISI 302; Goodfellow, UK) with an outside diameter 0.5 mm and an inside diameter 0.38 mm, was cut to three lengths: one at 15 mm to form the shaft and two 10 mm pieces to form the inlet and outlet tubes. The cut surfaces were filed with a watchmaker's broach to ensure all ends were smooth. Two 50 mm lengths of fused silica tubing (SGE Analytical Science, UK) with an outside diameter of 170 μm and an inside diameter of 110 μm were threaded inside the shaft with one threaded over the inlet and outlet tubes, respectively, to form a Y-shaped assembly. The end of the inlet silica tubing protruded from the end of the shaft by approximately 3 mm while the outlet tubing was drawn ~5 mm up inside the shaft. Hollow fiber cellulosic membrane (Filtral 12, AN69 HF, Hospal, 300 μm diameter) was cut to 10 mm and threaded over the silica tubing and inside the shaft. An epoxy adhesive (Araldite) was used to glue the microdialysis membrane to the steel with 3.5–4.0 mm protruding. A 0.5 mm epoxy plug was applied to seal the end of the membrane and epoxy was applied at the junction of the three steel tubes to hold the assembly together.

The probe was implanted vertically into the center of the exposed IC and held in place by a stereotaxic manipulator. Using a syringe pump, the probe was continuously perfused at 2 μl/min with artificial cerebrospinal fluid (aCSF: 140 mM NaCl; 3 mM KCl; 0.27 mM $Na_2H_2PO_4$; 1.2 mM $Na_2HPO_4$; 2.4 mM $CaCl_2$; 1 mM $MgCl_2$; and 7.2 mM Glucose). The pH of the aCSF was adjusted to 7.4.

To deactivate neuronal activity, the sodium channel blocker procaine (0.1 M, Sigma, UK) was dissolved in aCSF and infused through the probe. This concentration was selected as microdialysis probe efficacy is around 10% (**Boehnke and Rasmusson, 2001**). We estimate that this produced an infusion of ~2.5% procaine into the IC.

A switch between aCSF only and aCSF plus procaine was made at a junction 4 cm from the probe with the tubing clamped to prevent any mechanical disturbance.

## Stimulus Generation and presentation

Stimuli were generated digitally by Tucker-Davis Technologies System 2 (TDT2, Alachua, FL) hardware which was controlled by software that allowed the frequency and level of stimuli to be varied in real time. Pure tones were generated digitally and were $cosine^2$ ramped at the onset and offset. Search stimuli were 50-ms tone bursts, roved manually between 0.1 to 20 kHz and between 0 to 99 dB attenuation, and presented at a repetition rate of 4 Hz.

Stimuli were delivered through Sony (UK) MDR 464 earphones housed in an alloy enclosure and coupled to damped probe tubes (4 mm diameter) that fitted into the Perspex specula (**Rees et al., 1997**). The output of the system was calibrated using an eighth inch Bruel & Kjaer (UK) Type 4138 microphone, a Type 2639 preamplifier and Type 2610 measuring amplifier. The microphone was seated in a small tubing coupler that sealed the narrow end of the speculum holding each speaker. The maximum output of the system was approximately flat from 0.1 to 9 kHz (100 ± 8 dB SPL) and then fell with a slope so that

the maximum output of the system at 16 kHz was 78 dB from both the left and right speakers. Second and third harmonic components in the signal were approximately 60 dB of the fundamental at the highest output level. The majority of recordings were made in the flat region of the system below 10 kHz.

## Electrophysiological recordings

All recordings were obtained using borosilicate glass-coated tungsten electrodes. The electrode position was controlled by a stepper-motor microdrive activated by a remote control outside the sound-attenuating room. Electrode penetrations were made through the cortex in a mirror opposite position to the center of the exposed IC. Extracellular action potentials were amplified (×10,000) and band-pass filtered (0.1–3 kHz) by an amplifier (Dam-80; World Precision Instruments, UK). Spikes were further high pass filtered (300 Hz) via TDT2 hardware before being discriminated online, converted to logic pulses, and time stamped to an accuracy of 10 μs.

Peri-stimulus time histograms (PSTHs) were generated to sound-driven multi-unit activity with an electrode inserted adjacent to the microdialysis probe. The stimuli to drive these responses were 100 repetitions of a CF tone at 20 dB above threshold. Multiple PSTHs and FRAs were recorded before, during and following infusion of procaine.

In the IC contralateral to deactivation, only single units were recorded. On isolation of a single unit, the CF and minimum threshold to tones presented to the ear contralateral to the recorded IC were estimated to establish the settings for data collection. The parameters of the stimuli used to test each neuron were defined relative to this initial estimate.

FRAs were generated from the pseudorandom presentation of pure tones to binaural diotic stimuli (75 ms duration, 5 Hz repetition rate, 5 ms cosine$^2$ ramped rise/fall time) from two octaves above to three octaves below the estimated CF in 1/10th octave intervals. The stimulus level could be varied from 10 to 90 dB attenuation from the maximum output of the system in 5 dB steps. The number of spikes produced in response to each stimulus was counted and displayed at the appropriate position in a plot of tone frequency vs attenuation, according to a color scale.

RLFs were generated from responses to CF tones (75 ms duration, 4 Hz repetition rate) presented in a pseudorandom manner from 10 to 90 dB attenuation at a resolution of 10 dB. The number of spikes fired to each sweep was stored. RLFs were constructed from either 20 or 50 repetitions of each stimulus.

## Histological and image processing

At the conclusion of an experiment the animal was deeply anesthetized by an i.p. injection of sodium pentobarbital (Euthanal, Merial, UK, 200 mg/ml, 2 mls) and the animal was perfused transcardially with 300 ml of a wash solution comprised of 0.1 M phosphate buffered saline (PBS) at a pH of 7.4 (±0.1), followed by perfusion of 200 ml of a fixative solution (4% paraformaldehyde in PBS). After further fixation, the brain was cryoprotected in 30% sucrose in 4% paraformaldehyde in PBS solution until it sank before being embedded in Tissue-Tek (Sakura, NL) and frozen. The brain was sectioned at 60 μm in the coronal plane on a cryostat (HM 560, Microm, UK) and stained for Nissl substance with cresyl violet. Sections through the IC were imaged with an Axio Imager microscope running AxioVision software with a motorized stage (Zeiss). The MosaiX module was used to collect images from each section containing the entire IC. The perimeter of the IC and the lesion caused by the microdialysis probe were measured using the AxioVision outline module.

## Data analysis

Initial characterizations of each unit response to the FRA paradigm were designated as the control FRAs to which FRAs measured during and after contralateral IC deactivation were compared. In this regard, each neuron served as its own control. FRAs and RLFs were recorded throughout and following deactivations lasting less than 30 min. Units were allowed an hour to recover, although the majority of those that recovered did so within 30 min of terminating cooling or MDP.

Owing to the need to minimize the period of deactivation to ensure recovery recordings could be made, multiple FRAs could not be collected in each condition. For this reason statistical comparisons for individual units between conditions were not possible. However, because each FRA is made up of the response to many hundreds of stimuli, there are strong grounds to support the contention that each FRA is a reliable and repeatable representation of that unit's frequency response (*Palmer et al., 2013*). Furthermore recordings in control conditions show a high degree of repeatability for the analysis.

The CF and threshold were acquired from each FRA (*Ingham et al., 2006*) and responses to the lowest level of stimulation across all frequencies within the FRA were used to define the mean spontaneous firing rate. Those bins within each FRA that contained values greater than two standard deviations above the mean spontaneous rate were identified. For each frequency the lowest stimulus level at which the spike rate exceeded the mean spontaneous rate plus two standard deviations was deemed to be the excitatory threshold for that frequency. The array of excitatory thresholds was fitted with a tenth order polynomial which produced the excitatory frequency tuning curve (FTC) of the FRA. The minimum point of the FTC was used to define the CF and threshold of the unit. The number of frequency-level response bins within each FRA contained within this classifier were counted and compared between conditions. For some FRAs (n = 27), a tenth order polynomial was not accurate in reflecting the FRA. For these FRAs the number of bins exceeding a given number (1 spike per bin for most of these neurons) of spikes was counted. The number of spikes in each FRA was also counted and compared between conditions for each unit.

The half maximal firing rate of RLFs was measured by normalizing to the maximum firing rate in each condition and calculating the point along the abscissa which formed a right angle to the half firing rate on the ordinate.

ROC analyses for each unit involved the pairwise comparison of all adjacent stimulus values in the RLF. The area under ROC function for each comparison was used as a measure of discriminability between those two stimuli. The area under the curve for each pairwise comparison was summed to create the 'discriminability index' for each RLF.

We calculated a modulation index for the changes between control and deactivation conditions for numerous measures. Each modulation index was constructed in the standard manner using the equation: MI = (deactivated − control) / (deactivated + control).

## Statistical analyses

Non-parametric statistical tests have been used throughout. All reported p values are exact and two tailed. A two sample Kolmogorov-Smirnov test was employed to assess changes in FRA area and FRA spikes between the distributions of V-shaped and nonV FRAs. Friedman's repeated measures ANOVA on ranks was used in population analyses of changes in half maximal firing rate in RLFs and changes in ROC discriminability. For *post hoc* analyses, two Wilcoxon signed rank tests were performed: one between the control group and the deactivated group; the other between the control group and the recovery group. For these tests the α was Šidák corrected to 0.0253 because two *post hoc* tests were employed on each occasion. A Spearman rank correlation was used to investigate the relationship between changes in the half maximum firing level in the RLF and changes in threshold. Fisher's exact test, structured as a 5 × 2 (rows x columns) contingency table, was used to test changes in RLF type using a binary classification for each unit, with zero being no change and one being change. These two columns were summed for each of the five RLF types found in the dataset. We did not predetermine sample sizes as it was impossible to provide an accurate estimate of the effect of our experimental manipulation.

## Acknowledgements

We thank Sasha Gartside and Richard McQuade for help with establishing the microdialysis technique, and Trevor Booth from the Bio-Imaging Unit in the Faculty of Medical Sciences at Newcastle University for help with microscopy. We thank Tim Griffiths, Chris Petkov and Claudia Racca for their helpful comments on an earlier draft of the manuscript. This research was supported by BBSRC grant BB/J008680/1 to AR.

## Additional information

### Funding

| Funder | Grant reference number | Author |
| --- | --- | --- |
| Biotechnology and Biological Sciences Research Council | BB/J008680/1 | Adrian Rees |

The funder had no role in study design, data collection and interpretation, or the decision to submit the work for publication.

## Author contributions

LDO, AR, Conception and design, Acquisition of data, Analysis and interpretation of data, Drafting or revising the article

## Ethics

Animal experimentation: All experiments in this study were performed in accordance with the terms and conditions of a licence (PPL 60/3934) issued by the UK Home Office under the Animals (Scientific Procedures) Act of 1986, and with the approval of the Local Ethical Review Committee of Newcastle University. All procedures were performed under anesthesia, supplemented with analgesia.

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
