## [Decision Letter]

Thank you for sending your work entitled “The cooperative colliculi: midbrain commissural control of auditory processing” for consideration at *eLife.* Your article has been favorably evaluated by Eve Marder (Senior editor), 2 reviewers, and a member of our Board of Reviewing Editors.

The Reviewing editor and the reviewers discussed their comments before we reached this decision, and the Reviewing editor has assembled the following comments to help you prepare a revised submission.

The reviewers found your paper quite interesting by showing that inactivation of the IC affected the response properties of the other IC to sound stimulation. Your interpretation is that the ICs of both sides interact to processing sound stimuli. However, the most substantive critique made by the two reviewers concerned the inactivation protocols and the lack of information about the precise IC region that is inactivated. Ideally, there would be evidence of the area inactivated in each experiment. Reviewer #3 is not sure of the specificity of silencing the commisural fibers, and that some of the effects observed could be due to silencing also the dorsal nucleus of the lateral lemniscus (DNLL).

We hope that you can respond to these critiques with your current database. If this is possible, we would be happy to receive a revision, provided you respond to the reviewers' critiques. In addition, both reviewers commented on dissimilarity as an important function of sensory systems for representing sensory information. The reviewers found in the introduction section some statements for example, that the commissure of IC is the most prominent crossing of the brainstem auditory pathways. He/she notices the trapezoid body probably contains more crossing fibers and that it could be good to cite the series of papers on acoustic chiasm. Also, the reviewer would like you to cite recent papers on binaural responses in the IC from whole-cell in vivo responses, as they could be important for the data interpretation of this paper.

*Reviewer #2*:

Results: V-shaped monotonic FRAs were apparently unaffected by contra IC deactivation (Figure 2). Claimed, however, that firing rate changed in most FRAs. Quantitative data not shown.

Claimed that non-V FRAs showed a fundamentally different response, with changes in both area and firing rate resulting from contra IC deactivation (Figure 3). However, I'm unsure these are more than quantitatively different from those in Figure 2. All FRAs, however, were convincingly reversed to control levels following reactivation.

The only difference between the two neuron classes and, in fact, the effect of deactivation was to produce a much larger range of changes in FRA area for nonV FRAs than for V FRAs. This is to be expected given the likely broader field and type of contralateral influence on the dendritic trees of nonV compared with V units. This observation is interesting, but doesn't seem to be followed up by any detailed discussion.

I have difficulty with the claim that the two “non-monotonic” units in Figure 5 became “monotonic” during contra IC deactivation. The Unit in Figure 5 doesn't appear to have changed much (not significantly?), while that in Figure 5 became more insensitive and unresponsive rather than losing its non-monotonicity. Similarly, the “straight” response in Figure 5 could be simply insensitivity, with the non-monotonic portion of the response occurring at higher than tested sound levels. Given these issues, I find it difficult to accept that the input to the binary classifier was meaningful. Finally, I don't see the relation with just one ('closed') of the FRAs in Figure 3 when there were just 12 non-monotonic units in total whose RLF was analyzed, as I understand it.

The interpretation of the data in Figure 5 as signaling a change in sound level would normally be understood to mean a change in slope of the RLF, not a lateral shift in the whole RLF (Figure 5) which is more indicative of a general desensitization of the response, as above. This possibility is examined (l.117-124). It appears that the units shown in Figure 5 did indeed have a certain disconnect between threshold and half-maximal firing rate, in that firing appeared to increase at a lower level (threshold) than expected from the main ascending arm of the RLF. But the observation from these units is that the main ascending arm of the RLF had a slope very similar to that of the same units before and after contra inactivation. If the units in 5b,c are unrepresentative, they should be replaced, but they do serve as examples of how a disconnect between threshold and half-maximal firing may not be a good metric of overall sensitivity to sound level.

The interesting ROC analysis shows that, on average, IC neurons (with non-monotonic RLFs? unclear, but Figure 6 suggests it may be all 38 isolated neurons) had slightly less sensitivity to changing sound level when the contra IC was deactivated. Although statistically significant, this effect is small, and I wonder how functionally meaningful it is.

Discussion:

The notion that spectral processing may be other than a monaural (presumably binaural?) phenomenon would only be borne out of these data if it could be shown that it required binaural stimulation. That both sides of the brain are involved in almost any sound processing is hardly new. Although sub-cortical auditory connections go through essentially two 'chiasma', one at the connection from CN to SOC and another at connection from SOC to IC, it is well understood that there are (admittedly weaker) ipsilateral as well as contralateral connections throughout. Stimulating a single ear can therefore activate both the contralateral and ipsilateral connections. I don't think that this process could be considered anything other than 'monaural', even though diotic stimulation was used in this study.

Does the suggested continuum of nonV responses extend also to the V category? In other words, could so-called V responses be just an extreme version of the nonV continuum? This seems a rather important issue in light of the separation throughout this paper between V and nonV RFAs, and the somewhat ambiguous nature of many of the RFAs and RLFs shown in the paper.

Explain how the “representations of auditory space” observed in the IC are not “fully formed” at the level of SOC. I think this brief description sounds more dramatic than is the actual situation. It also implies some sort of map of auditory space in the IC which does not, in fact, exist.

*Reviewer #3*:

Results:

The authors should report the numbers of V-type neurons that had increased, decreased, or unchanged firing rates.

Firing rate numbers should also be reported for nonV type neurons. These data might be combined since there is not difference in the amount of change for the two types.

The authors should briefly describe the criteria used to classify Rate level functions (RLF). This is important since the data for Figure 5 describes changes from non-monotonic to other types. It is not clear that Figure 5 shows neurons changing from non-monotonic to monotonic. Figure 5 appears to be saturating and 5c remains non-monotonic.

Discussion:

To what extent have monaural and binaural FRAs been compared in the same neuron?

The authors should be aware that there is no anatomical evidence of GABAergic or other interneurons in the IC. It is suspected but unproven that GABAergic IC neurons have local axonal collaterals in addition to axons that project out of the IC.

The discussion of the contribution of the binaural system to spectral coding is not straightforward. Mechanisms relevant to combining information from two hemifields should be discussed in the context of the present results.

---

## [Author Response]

*The most substantive critique made by the two reviewers concerned the inactivation protocols and the lack of information about the precise IC region that is inactivated. Ideally, there would be evidence of the area inactivated in each experiment. Reviewer #3 is not sure of the specificity of silencing the commisural fibers, and that some of the effects observed could be due to silencing also the dorsal nucleus of the lateral lemniscus (DNLL)*.

We agree we should have included more detail on the deactivation protocols. We do reference our previous paper (44), which is devoted to validating the cooling technique in our preparation, but we should have summarized the major points in the current paper. These points have now been added to emphasize that the measured deactivation of the IC extends to depths representing frequencies up to ∼8 kHz, i.e. the dorsal half of the IC. On this basis we believe that the DNLL, which lies below the IC, was not deactivated. In addition we took a number of steps to avoid the possibility of the spread of cooling, including limiting the time for which cooling was applied. For practical reasons it was not possible to make these measurements in every experiment, but our measures in numerous control experiments reported in [44] give us confidence that the extent of cooling is within the same bounds from experiment to experiment. Our use of microdialysis of procaine was to provide an additional control for the spread of cooling, although the effects reported with this method are generally more modest because the spread of the effect is more spatially restricted within the IC. We do have confirmation from each MDP experiment that the lesion from the microdialysis probe was always confined within the left IC. We can therefore unequivocally rule out deactivation of the DNLL in these experiments and we found similar effects to cooling experiments in these experiments. Thus, while we cannot exclude the possibility that the DNLL was affected in every cooling case, we believe it is highly unlikely. We do cover these points in the Discussion, but agree that appropriate statements were needed in the Methods. We have added text to the Methods and the Results to clarify these issues.

*Both reviewers commented on dissimilarity as an important function of sensory systems for representing sensory information*.

We thank the reviewers for raising the issue of dissimilarity in the representation of sensory information. We have added to our discussion of this issue to emphasize the implications of our data in light of this important point.

*The reviewers found in the introduction section some statements for example, that the commissure of IC is the most prominent crossing of the brainstem auditory pathways. He/she notices the trapezoid body probably contains more crossing fibers and that it could be good to cite the series of papers on acoustic chiasm. Also, the reviewer would like you to cite recent papers on binaural responses in the IC from whole-cell in vivo responses, as they could be important for the data interpretation of this paper*.

We thank this reviewer for this helpful clarification and have modified the text to include more introductory comments about the acoustic pathways in the brainstem and on recent papers that reported whole-cell in vivo responses in the IC and how they pertain to our data. Our point here is that commissure of the IC connects two homologous structures which contrasts to the acoustic chiasm in which fibers cross the midline but connect nuclei higher up the pathway rather than at the same level. We have now made the Introduction more comprehensive and cited the papers requested by the reviewers.

Reviewer #2:

*Results: V-shaped monotonic FRAs were apparently unaffected by contra IC deactivation (*Figure 2*). Claimed, however, that firing rate changed in most FRAs. Quantitative data not shown*.

This point highlights an omission of a signpost to Figure 4 where we present the data quantifying this claim. We have now inserted a reference to that figure to signpost where we have quantified this effect. We have also now included quantitative data in the text to show the changes in firing rate.

*Claimed that non-V FRAs showed a fundamentally different response, with changes in both area and firing rate resulting from contra IC deactivation (*Figure 3*). However, I'm unsure these are more than quantitatively different from those in*
Figure 2*. All FRAs, however, were convincingly reversed to control levels following reactivation*.

As with the previous comment, we have now inserted an earlier reference to Figure 4 and d, to highlight the quantitative analyses showing that the changes in area of non-V-shaped FRAs were greater than in V-shaped FRAs.

*The only difference between the two neuron classes and, in fact, the effect of deactivation was to produce a much larger range of changes in FRA area for nonV FRAs than for V FRAs. This is to be expected given the likely broader field and type of contralateral influence on the dendritic trees of nonV compared with V units. This observation is interesting, but doesn't seem to be followed up by any detailed discussion*.

We respectfully disagree with the reviewer’s assertion that nonV shaped FRAs likely have broader dendritic fields. There is no evidence that we are aware of supporting this view. Indeed, [61], who recorded and reconstructed single IC neurons using juxtacellular methods, found V shaped FRAs for units with the largest and smallest dendritic trees in their sample; thus demonstrating no relationship between dendritic area and FRA type. One of the puzzles of the IC is this lack of obvious correspondence between morphology and physiology. Such evidence may yet come to light, but more data is required to investigate this issue. Previous studies (e.g. [30]; [46]) do suggest that units with V and nonV responses receive different inhibitory input, and our current findings suggest that intercollicular connections account, at least in part, for these differences.

*I have difficulty with the claim that the two “non-monotonic” units in*
Figure 5
*became “monotonic” during contra IC deactivation. The Unit in*
Figure 5
*doesn't appear to have changed much (not significantly?), while that in*
Figure 5
*became more insensitive and unresponsive rather than losing its non-monotonicity. Similarly, the “straight” response in*
Figure 5
*could be simply insensitivity, with the non-monotonic portion of the response occurring at higher than tested sound levels. Given these issues, I find it difficult to accept that the input to the binary classifier was meaningful. Finally, I don't see the relation with just one ('closed') of the FRAs in*
Figure 3
*when there were just 12 non-monotonic units in total whose RLF was analyzed, as I understand it*.

We classified non-monotonic RLFs using the criteria of [49]. Classification as a non-monotonic RLFs required a 25% reduction in firing rate at higher sound levels than maximum firing rate. The unit in Figure 5 had a reduction of 30% in firing rate at higher sound levels in the control condition, and was therefore classified as non-monotonic. During procaine infusion, the largest reduction from peak firing was 5%, and was therefore reclassified as being monotonic saturating. We interpret the change in Figure 5 as being a reduction in both the sensitivity to lower sound levels and a change in monotonicity. While the unit showed reduced sensitivity to lower sound levels, an increase in firing rate still occurred over 50 dB, indicating the unit could still signal changes in sound level. In the control condition the unit reduced by 55% from maximum firing and was classified as being non-monotonic. During deactivation this became a reduction of 14% during deactivation as the RLF was reclassified as being monotonic saturating. This supports our classifications of a change from non-monotonic to monotonic across the wide range of stimuli we presented. We accept the criticism that the changes in the unit in Figure 5 do not preclude the possibility that the non-monotonic portion of the response could have occurred at higher sound levels than those presented. We make the claims of changes in non-monotonicity only across the range of stimuli we presented, as it would be impossible to know what the response of any unit would be to stimuli that were not presented. We prefer to retain this example as a good example of a change in sensitivity. We accept there are too few examples of closed FRAs to support this point and we have removed this comment from the manuscript.

*The interpretation of the data in*
Figure 5
*as signaling a change in sound level would normally be understood to mean a change in slope of the RLF, not a lateral shift in the whole RLF (*Figure 5*) which is more indicative of a general desensitization of the response, as above. This possibility is examined (l.117-124). It appears that the units shown in*
Figure 5
*did indeed have a certain disconnect between threshold and half-maximal firing rate, in that firing appeared to increase at a lower level (threshold) than expected from the main ascending arm of the RLF. But the observation from these units is that the main ascending arm of the RLF had a slope very similar to that of the same units before and after contra inactivation. If the units in 5b,c are unrepresentative, they should be replaced, but they do serve as examples of how a disconnect between threshold and half-maximal firing may not be a good metric of overall sensitivity to sound level*.

We appreciate the reviewer drawing attention to the nuanced dissociation between the shifts in half maximum firing rate and the changes in discrimination due to changes in the slopes of the RLFs. The analyses we present in in Figure 5 are the quantification of the change in supra-threshold sensitivity of the population of 38 units. We accept that this analysis does not directly pertain to a change in discriminability that we found and quantified in Figure 6. We have changed the heading for this section to make this clear. The example units we present in Figure 5 were chosen to exemplify the change in half maximum firing rate that was found across the population of 38 units, and as such, we believe that the examples we present are appropriate within this figure to support this finding. While this section quantifies the increase in half maximum firing rates, we have determined that the dynamic spiking range of each example unit in Figure 5 reduced during deactivation. While these units are not the best examples of this effect, they contribute to the population data outlined in the following section that quantifies the reduction in discriminability.

*The interesting ROC analysis shows that, on average, IC neurons (with non-monotonic RLFs? unclear, but*
Figure 6
*suggests it may be all 38 isolated neurons) had slightly less sensitivity to changing sound level when the contra IC was deactivated. Although statistically significant, this effect is small, and I wonder how functionally meaningful it is*.

The reviewer is correct that this analysis was applied to all 38 units in our sample and we have clarified this in the text. The analysis shows that the ability of the population of IC neurons to signal changes in sound level was reduced during deactivation. The reviewer interprets the effect as small; however this may be due to the analysis method employed. For example, the dynamic spiking range in the example neuron in Figure 6 reduced by 45% compared to the control condition. The ability of an ideal observer to decode changes in sound level from a change in firing rate is dependent on the variance of the responses as well as the dynamic range and gradient of the slope, something out DI measure takes account of. In the example unit, the 45% reduction in dynamic range of the slope was quantified as a 20% reduction in DI. This indicates that while DI quantified the changes observed, accounting for the variance in the response as well as the slope lost some sensitivity. That being said, it is to our knowledge, the first and only method to be able to quantify the changes in neural discrimination across the range of stimuli presented. Using this method, we found a >10% reduction in median and inter-quartile range of population DI during deactivation. We would argue that reducing the ability of the population of IC neurons to encode changes in sound level by >10% represents a significant contribution of CoIC mediated processing to this task. Indeed, that lateral projections contribute at all to this task is a novel and unexpected finding.

*Discussion*:

*The notion that spectral processing may be other than a monaural (presumably binaural?) phenomenon would only be borne out of these data if it could be shown that it required binaural stimulation. That both sides of the brain are involved in almost any sound processing is hardly new. Although sub-cortical auditory connections go through essentially two 'chiasma', one at the connection from CN to SOC and another at connection from SOC to IC, it is well understood that there are (admittedly weaker) ipsilateral as well as contralateral connections throughout. Stimulating a single ear can therefore activate both the contralateral and ipsilateral connections. I don't think that this process could be considered anything other than 'monaural', even though diotic stimulation was used in this study*.

We accept this critique and have modified the discussion to reflect this. We would argue that our data do represent evidence of bilateral enhancement of spectral processing within the brain; however, this is different from binaural and has been removed from the text.

*Does the suggested continuum of nonV responses extend also to the V category? In other words, could so-called V responses be just an extreme version of the nonV continuum? This seems a rather important issue in light of the separation throughout this paper between V and nonV RFAs, and the somewhat ambiguous nature of many of the RFAs and RLFs shown in the paper*.

We agree that all FRAs in IC do reside at some point on a continuum of shapes; however V-shaped FRAs are a special category, as they are the only response type present in auditory nerve fibers and are preserved from the periphery to the cortices. There is considerable evidence that nonV shaped FRAs are formed by modification of V-shaped FRAs (e.g. [30], [46]); therefore V FRAs can be thought of as the spectral-level prototypes of the auditory pathway. Although the nonV FRA group contains several different types, they are alike in being shaped to some extent by inhibition and thus the distinction between V and nonV is not arbitrary. Furthermore, V and nonV FRAs can be categorized as distinct from each other using both objective and subjective methods (Palmer et al., 2103) and have been reported by all investigators into spectral processing in the IC. We believe it is therefore robust to classify V-shaped FRAs in one category and non-V shaped FRAs in another.

*Explain how the “representations of auditory space” observed in the IC are not “fully formed” at the level of SOC. I think this brief description sounds more dramatic than is the actual situation. It also implies some sort of map of auditory space in the IC which does not, in fact, exist*.

It was not our intention to imply that there may be a map of auditory space in the ICs. Our point is that while the location of sounds in auditory space in the azimuthal plane are first computed in the SOC, the representations of ITD and ILD cues are largely processed separately in the MSO and LSO respectively. ILDs are further processed in complex brainstem circuitry involving the DNLL before reaching the ICs. Additionally, the spectral cues required for localization in the vertical plane are processed in the DCN before reaching the ICs. The point we wanted to make is that the first point in the auditory pathway where all three of these cues converge and might interact is in the ICs. We have amended the text to clarify this point.

Reviewer #3:

*Results*:

*The authors should report the numbers of V-type neurons that had increased, decreased, or unchanged firing rates*.

*Firing rate numbers should also be reported for nonV type neurons. These data might be combined since there is not difference in the amount of change for the two types*.

We have modified the text to show these data and to direct the reader to Figure 4 and b, which show data for each unit in the sample from which we quantified these data.

*The authors should briefly describe the criteria used to classify Rate level functions (RLF). This is important since the data for*
Figure 5
*describes changes from non-monotonic to other types. It is not clear that*
Figure 5
*shows neurons changing from non-monotonic to monotonic.*
Figure 5
*appears to be saturating and 5c remains non-monotonic*.

We defined non-monotonicity according to the criteria set out in [49]. Non-monotonic responses were classified as those which had a 25% or greater reduction in firing rate at higher sound levels than the maximum firing rate. Using this criterion, the units in Figure 5 a-c were all defined as non-monotonic in the control condition. All lost that non-monotonicity during deactivation. Non-monotonic RLFs were the only type of RLF to have any change in RLF classification during deactivation. We have added more detail on our classification method.

*Discussion*:

*To what extent have monaural and binaural FRAs been compared in the same neuron*?

We have recorded FRAs to contralateral as well as binaural stimuli; however all analyses presented in this study are from binaural stimuli alone.

*The authors should be aware that there is no anatomical evidence of GABAergic or other interneurons in the IC. It is suspected but unproven that GABAergic IC neurons have local axonal collaterals in addition to axons that project out of the IC*.

We agree this was poor usage, and rather than using ‘interneurons’ we now refer to poly-synaptic local circuits that may be mediated by many sources.

*The discussion of the contribution of the binaural system to spectral coding is not straightforward. Mechanisms relevant to combining information from two hemifields should be discussed in the context of the present results*.

We appreciate this comment as it highlights an area of the Discussion that was unclear to the reviewer and we have enhanced the Discussion. We have made cautious modifications to the text as some of the issues raised as a consequence of this study are speculative.